# Atomic Databases: Four of a Kind

**Claudio Mendoza** [1,2] 

1    Department of Physics, Western Michigan University, Kalamazoo, MI 49008, USA;
     claudio.mendozaguardia@wmich.edu
2    Physics Center, Venezuelan Institute for Scientific Research (IVIC), Caracas 1020, Venezuela

**Abstract:** In the context of atomic data computations for astrophysical applications, we review four different types of databases we have implemented for data dissemination: a database for nebular modeling; TIPTOPbase; OPserver; and AtomPy. The database for nebular plasmas is briefly discussed as a study case of a successful project. TOPbase and the OPserver were developed during the Opacity Project, an international consortium concerned with the revision of astrophysical opacities, while TIPbase was part of the Iron Project to calculate radiative transition probabilities and electron impact excitation collision strengths for iron-group ions. AtomPy is a prototype for an open, distributed data-assessment environment to engage both producers and users. We discuss design strategies and implementation issues that may help in the undertaking of present and future scientific database projects.

**Keywords:** atomic databases; astrophysical applications; web services; data evaluation; Opacity Project; Iron Project

---

## 1. Introduction

Since the mid 1970s I have been involved in the calculation of atomic data for astrophysical applications, a specialized research field relevant to the spectral modeling of the plasmas associated with the wide variety of astronomical objects currently observed. By means of powerful terrestrial and space telescopes, the observable electromagnetic spectral windows now span from the radio to the $\gamma$ rays with unprecedented spectral and spacial resolution and sensitivity, and as a result of extensive sky surveys such as the Sloan Digital Sky Survey (SDSS[1]), we have rich spectra for several million objects. The atomic data demands in this astronomical big-data enterprise are consequently huge, not only in accuracy and completeness but also in access modes; therefore, database-centric computing has become established as a new paradigm [1].

If the computing of atomic data is in itself a life-time dedication, the design and implementation of efficient online database management systems (DBMS) require engineering skills originally alien to CPU-based scientific computing, which in most situations involve steep learning curves in research environments driven by fast changing information and communications technologies (ICT). Furthermore, the end products are not always warmly received by the data-user communities, and their long-term maintenance and upgrade are underfunded compromising sustainability. Despite such deterrents, the growth and diversity of distributed data repositories since the 1970s, boosted in the early 1990s by the emergence of the World Wide Web, have given rise to an unprecedented data deluge [2]. To illustrate this diversity I review four different case studies—four of a kind—I have

---

1    https://www.sdss.org/.

been involved with in order to highlight important issues to consider in the design, implementation, and maintenance stages of scientific databases.

## 2. Atomic Database for Nebular Modeling

In August 1982 the IAU Symposium 103 on Planetary Nebulae was held at UCL. Having defended my PhD thesis in this institution in March 1980, my supervisor, Mike Seaton, asked me to present an invited talk at this meeting on the advances of atomic calculations and experiments relevant to the study of these astronomical bodies. He additionally suggested including a selected and critically evaluated database of the atomic parameters—namely, level energies, radiative transition probabilities (*A*-values), electron impact collision strengths, and photoionization cross sections—used to model the forbidden and recombination lines observed in nebular plasmas. This was quite a task for a fledgling postdoc: it took me around twelve month to complete it under the ever stressful pressure of the impending deadline.

For such plasmas the computation of the required atomic data must take into account electron correlation effects (series perturbations and resonances) and relativistic effects (fine structure), which in the early days were treated with very approximate numerical methods [3–6]. These coveted datasets were compiled in the seminal treatise, *The Physics of Gaseous Nebulae*, by Osterbrock [7] and widely used in nebular modeling. The access to powerful computers in the 1970s led to a new generation of structure and electron–ion scattering calculations, which took formally into account electron correlation effects by the configuration interaction [8–10] and close-coupling [11–14] methods. Relativistic effects were introduced algebraically, with the Breit–Pauli Hamiltonian, or with the fully relativistic Dirac Hamiltonian. The new data volume and noticeable discrepancies with the early atomic parameters caused considerable distress in the nebular modeling community, which we intended to dispel with the publication of a recommended atomic database.

Both the review and database appeared in the proceedings of the IAU meeting [15] to become a highly cited paper, the database also being included in the book, *Physics of Thermal Gaseous Nebulae (Physical Processes in Gaseous Nebulae)*, by Aller [16]. The database essentially comprised 16 pages of flat tables organized in a manner practical to nebular modelers. As shown in Figure 1 for the iconic carbon isoelectronic sequence, it listed, in contrast to the previous compilation [7], effective collision strengths as a function of temperature. I made an attempt to ensure completeness although not all the ionic species, particular those of the third row ($11 \leq Z \leq 18$), had been studied with the new numerical methods; thus, the accuracy level of the compilation was not homogeneous, but the new database allowed modelers to share a reliable common atomic database that soon became standard reference.

What then makes a successful atomic database? This query was discussed in [17] adopting this database as a study case, which led both data producers and users to conclude that its acceptance was not a direct consequence of its completeness, accuracy, or regular updating but of the following precepts:

- The development of an atomic database must address the needs of prospective users;
- The publication of the database has to be timely;
- The compilation must become standard reference.

We must add that subsequent attempts to compile a more complete atomic database for emission-line diagnostics in nebulae (see, for instance, [18–20]) did not meet with a comparable reception until the appearance of CHIANTI[2] in the late 1990s [21]. In opinion this was due to slow piecemeal improvements and the reluctance of users to replace standard reference data. The widely shared atomic database allowed the nebular modelers to concentrate on the astrophysics rather than on the uncertainties of the underlying physical data. Rather than an atomic database,

---

[2]   https://www.chiantidatabase.org/.

CHIANTI is an application for modeling plasma emission lines which, although developed by the solar physics community, also includes nebular emission lines. It was originally coded in IDL—a popular but proprietary scripting language to analyze and visualize large scientific datasets (a Python version, ChiantyPy, is now available)—and requires local installation, but its well-honed functionality and regular database maintenance (see last update in [22]) have made CHIANTI a standard and sustainable enterprise.

| ION | PARM | $T_e(10^4 K)$ | $^1D_2-^3P_0$ | $^1D_2-^3P_1$ | $^1D_2-^3P_2$ | $^1S_0-^3P_1$ | $^1S_0-^3P_2$ | $^1S_0-^1D_2$ | $^3P_1-^3P_0$ | $^3P_2-^3P_0$ | $^3P_2-^3P_1$ | $^5S^o_2-^3P_1$ | $^5S^o_2-^3P_2$ |
|---|---|---|---|---|---|---|---|---|---|---|---|---|---|
| $c^0$ | ΔE | | 10192.6 | 10176.2 | 10149.2 | 21631.6 | 21604.6 | 11455.4 | 16.4 | 43.4 | 27.0 | 33718.8 | 33691.8 |
| | A | | 7.77−8 | 8.21−5 | 2.44−4 | 2.71−3 | 2.00−5 | 5.28−1 | 7.93−8 | 1.71−14 | 2.65−7 | 6.94 | 1.56+1 |
| | T | 0.05 | | 0.0625 | | | 0.0172 | 0.0620 | | | | 0.150 | |
| | | 0.1 | | 0.125 | | | 0.0339 | 0.0877 | | | | 0.212 | |
| | | 0.5 | | 0.603 | | | 0.149 | 0.196 | | | | 0.475 | |
| | | 1.0 | | 1.14 | | | 0.252 | 0.277 | | | | 0.671 | |
| | | 1.5 | | 1.60 | | | 0.320 | 0.340 | | | | 0.822 | |
| | | 2.0 | | 1.96 | | | 0.365 | 0.392 | | | | 0.950 | |
| $N^+$ | ΔE | | 15316.2 | 15267.5 | 15185.4 | 32640.1 | 32558.0 | 17372.6 | 48.7 | 130.8 | 82.1 | 46735.9 | 46653.8 |
| | A | | 5.35−7 | 1.01−3 | 2.99−3 | 3.38−2 | 1.51−4 | 1.12 | 2.08−6 | 1.16−12 | 7.46−6 | 4.8+1 | 1.07+2 |
| | T | 0.5 | | 2.64 | | | 0.352 | 0.405 | | | | 1.27 | |
| | | 1.0 | | 2.68 | | | 0.352 | 0.411 | | | | 1.28 | |
| | | 1.5 | | 2.72 | | | 0.359 | 0.418 | | | | 1.29 | |
| | | 2.0 | | 2.73 | | | 0.365 | 0.425 | | | | 1.27 | |
| $O^{2+}$ | ΔE | | 20273.3 | 20160.1 | 19967.1 | 43072.5 | 42879.5 | 22912.4 | 113.2 | 306.2 | 193.0 | 60211.8 | 60018.8 |
| | A | | 2.74−6 | 6.74−3 | 1.96−2 | 2.23−1 | 7.85−4 | 1.78 | 2.62−5 | 3.02−11 | 9.76−5 | 2.12+2 | 5.22+2 |
| | T | 0.5 | | 2.02 | | | 0.248 | 0.516 | 0.517 | 0.257 | 1.22 | 1.05 | |
| | | 1.0 | | 2.17 | | | 0.276 | 0.617 | 0.542 | 0.271 | 1.29 | 1.18 | |
| | | 1.5 | | 2.30 | | | 0.299 | 0.638 | 0.553 | 0.281 | 1.32 | 1.22 | |
| | | 2.0 | | 2.39 | | | 0.314 | 0.634 | 0.556 | 0.288 | 1.34 | 1.24 | |
| $Ne^{4+}$ | ΔE | | 30291.5 | 29879.1 | 29181.4 | 63501.2 | 62803.5 | 33622.1 | 412.4 | 1110.1 | 697.7 | 87950.7 | 87253.0 |
| | A | | 2.37−5 | 1.31−1 | 3.65−1 | 4.21 | 6.69−3 | 2.85 | 1.28−3 | 5.08−9 | 4.59−3 | 2.37+3 | 6.06+3 |
| | T | 0.5 | | 1.70 | | | 0.284 | 0.581 | 0.244 | 0.122 | 0.578 | 1.19 | |
| | | 1.0 | | 1.78 | | | 0.248 | 0.518 | | | | 1.51 | |
| | | 1.5 | | 1.85 | | | 0.240 | 0.550 | | | | 1.53 | |
| | | 2.0 | | 1.92 | | | 0.238 | 0.602 | | | | 1.51 | |

**Figure 1.** Excerpt from Table 6 of [15] showing the atomic database for the carbon isoelectronic sequence.

## 3. TIPTOPbase

The term TIPTOPbase—alluding to the adjective "tip-top" for the very best class and quality—refers to the TOPbase and TIPbase atomic databases of the Opacity Project (OP[3]) and Iron Project (IP[4]).

In the early 1980s a request was made, a plea in fact, for a revision of the astrophysical opacities due to inconsistencies in stellar evolution and pulsating theories [23]. This challenge was taken by two teams: the OPAL[5] group from the Lawrence Livermore National Laboratory and the international OP consortium of which I was a member. After a decade of intense computations, the opacities from these two projects were in surprisingly good agreement in spite of their different approaches to represent the equation of state and quantum mechanical frameworks to calculate the atomic radiative data [24]. Regarding the latter, the OP insisted in implementing state-of-the-art computational methods—namely, the *R*-matrix method [11] based on the close-coupling formalism—to account for electron correlation effects. As a result an atomic dataset of extraordinary volume (~1 GB) and accuracy was generated containing energy levels with principal quantum numbers $n \leq 10$, oscillator strengths ($gf$-values), and photoionization cross sections of both ground and excited states for cosmic abundant ions (atomic number $1 \leq Z \leq 26$ and electron number $1 \leq N \leq Z$) [25].

The OP was in fact a pioneer of what is now referred to as *collaborative big-data science* [1]. The workload was divided into isoelectronic sequences that were assigned to the respective research groups. Progress was monitored on a six-monthly basis in OP meetings held in the different participating countries and, in the latter stages of the project, through an email list. The atomic data compilation by means of half-inch magnetic tapes and exabyte cartridges was coordinated

---

3     http://cdsweb.u-strasbg.fr/topbase/TheOP.html.
4     http://cdsweb.u-strasbg.fr/topbase/testop/TheIP.html.
5     https://opalopacity.llnl.gov/opal.html.

by Mike Seaton himself, who devised a series of utilities to test data integrity leading in several cases to recalculations or new calculations (e.g., the PLUS-data [26]). I implemented a second set of tests for TOPbase mainly concerned with term assignments and spectroscopic series accuracy and completeness. Publication[6] was carried out in series of papers ("The equation of state for stellar envelopes", "Atomic data for opacity calculations") and in two books [27].

My contribution to the OP was mainly carried out while I was a scientific consultant at the IBM Venezuela Scientific Center in Caracas. Due to the large volume of data being produced, I was in a convenient place to develop an efficient DBMS to facilitate manipulation and access modes (interactive and batch) of the new OP atomic datasets, a computational tool most members of our scientific community of data producers and users was not familiar with. Since commercial DBMSs were out of the question due to price and portability issues, we developed from scratch the command-based DBMS in standard Fortran 77 of what came to be known as TOPbase[7] [28]. Database distribution and access modes were also seriously pondered at the time between periodic CD-ROM releases or, alternatively, remote access from a central site through the TCP/IP `telnet` application protocol on the new UNIX scientific workstations. We fortunately chose the latter, and with the fast advent and ubiquitous expansion of the Internet and World Wide Web, TOPbase was in fact ahead of its time.

Another important aspect in the rise of online scientific database services was the emergence of data centers, among which the strategic alliance of the OP with the CDS[8] was key in guaranteeing the TOPbase long-term service quality, data integrity, and security [29]. It is worth mentioning that the original TOPbase at the CDS is still operable, but was recently transferred to a MySQL[9] DBMS by Franck Delahaye and Nicolas Moreau (Observatoire de Paris, France) to integrate it to the portal of the Virtual Atomic and Molecular Data Center (VAMDC[10]), an ambitious European project to integrate several (more than 30) atomic and molecular databases [30,31].

As discussed in [28], TOPbase manipulates data associated with ionic states (term energies and photoionization cross sections) and dipole allowed transitions ($gf$-values), and as shown in Figure 2, its file structure comprises two set of indexes ($e$ and $f$ indexes) and three datasets: $e$ containing term energies, $f$ containing $gf$-values, and $p$ with photoionization cross sections, the latter being the more voluminous (90%) as it contained lengthy energy tabulations. When a logical data search is requested, the indexes are loaded into main memory to fetch directly the data from the bulk of the database. Indexes are structured to: (i) provide a table of contents; (ii) reduce a single search to one disk access in the $e$ and $f$ entities and to two disk accesses in $p$; and to optimize multiple searches. The index structure in TOPbase ensures efficient searches along isoelectronic and isonuclear sequences and the fast sorting of energy levels and transition wavelengths. The main system performance limitation is data uploading from disk, a process that is accelerated by storing data under the random-access binary format.

The TOPbase functional design is shown in Figure 3, which can certainly be used as a general template in atomic and molecular database design. It emphasizes data compactness and fast access by managing main and secondary storages jointly through two data structures in main memory: the *view* and the *table*. The *view* is a database subset resulting from a search specified by the user selected criterion referred to as the *view descriptor*. The *table* structure allows further logical reorganizations of the *view* (e.g., sorting, column and row exclusions, etc.) to satisfy the user output requirements, which can then be finally directed to a monitor, printer, or disk. In the TOPbase web-based version, the *view* and *table* are both specified in the HTML query form and reduced to a single event rather than an iterative sequence as in the command-based version. Furthermore, the DBMS stores on disk an

---

6　　http://cdsweb.u-strasbg.fr/topbase/publi.html.
7　　http://cdsweb.u-strasbg.fr/topbase/topbase.html.
8　　http://cdsweb.u-strasbg.fr/.
9　　https://www.mysql.com/.
10　https://portal.vamdc.eu/vamdc_portal/.

active log of *view descriptors* summarizing the user search activities to expedite subsequent searches and to keep an abridged search record; i.e., selected *view descriptor* entries can be excluded or all completely erased. The TOPbase command-based version also allowed graphic displays of table columns and cross sections, which in the web-based are upgraded to interactive cross-section plots by means of Java applets.

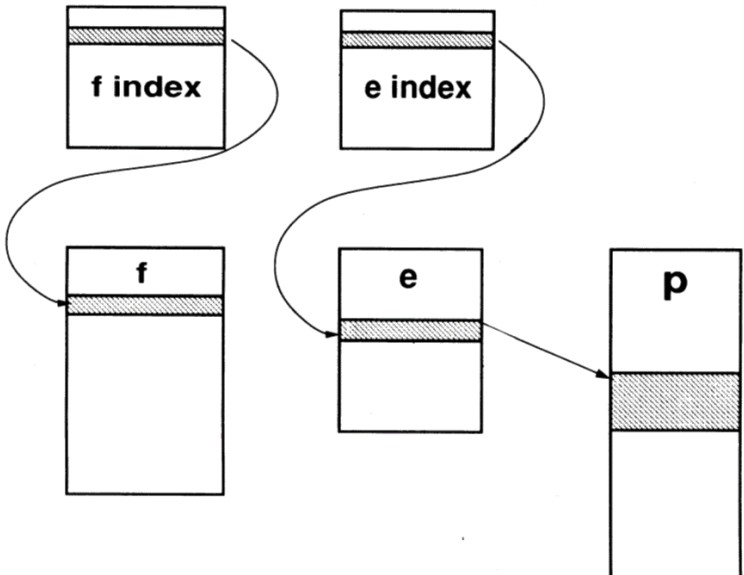

**Figure 2.** TOPbase file structure showing the *e*, *f*, *p* datasets and the *e* and *f* indexes. Reproduced from Figure 1 of [28].

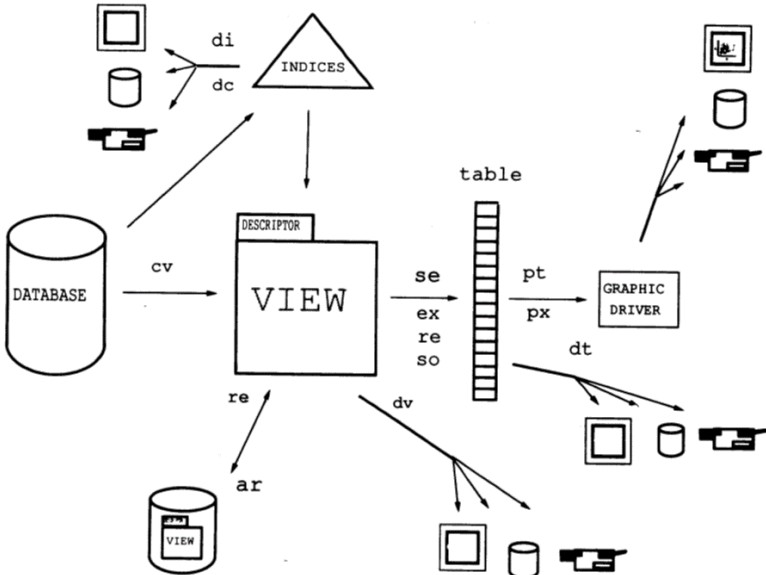

**Figure 3.** TOPbase functional blueprint showing: the two main data structures, the *view* and the *table*; the display, printing, and graphic capabilities; and the query commands of the original command-based version. Reproduced from Figure 2 of [28].

As the OP computations of the atomic radiative data came to an end in the mid 1990s, we soon embarked on a second big-data collaboration, the IP, to compute radiative and collisional data for iron-group ions [32]. Following the favorable outcome of TOPbase, the natural step was to

adapt its DBMS to handle the new volume of data bearing similar characteristics, namely TIPbase[11]. Although TIPbase is currently operational at the CDS, it is of little value as it does not contain most of the datasets computed in the IP, which were directly transferred to the CHIANTI[12] application to compute level populations for modeling emission lines in the solar corona and flares [21,22].

Two important points in scientific database management are illustrated with the fate of TIPbase. One is the database functional level required by prospective users, the higher the better, as data transcription to and maintenance in a modeling code is usually an involved process. Therefore, due to the large data volumes being generated in the current data era, scientific computing is rapidly becoming database centric; i.e., most applications, tools, and utilities are run where the databases reside rather than at the user end. The second is the competition between data producers and collectors in the context of data provenance, where the producer is often obviated despite *bona fide* efforts by the collector to request users to quote the original sources. TIPbase was conceived only to display data computed in the IP and could not then compete in completeness with CHIANTI that compiled data from different sources.

## 4. OPserver

Since the ubiquitous inception of the World Wide Web in the 1990s, most scientific databases are now accessed interactively through web pages. However, as mentioned in Section 3, there is also the need for batch access; i.e., for direct application-to-application interoperability. When we initially considered efficient access modes to the OP opacities, we soon arrived at the concept of an application web server rather than a database, namely the OPserver[13] [33], which would allow different access modes and fast response to wide user demands regarding chemical mixtures and thermodynamic conditions (temperature and density). In contrast, OPAL provided access to tables of opacity means pre-computed for selected chemical mixtures and temperature–density grids for users to interpolate locally to suit their needs.

Astrophysical opacities are usually required in the form of opacity means; e.g., Rosseland and Planck means that imply weighted integration of voluminous tabulations of monochromatic opacities as a function of photon frequencies. Consequently, the lengthy data readings from disk are the main overhead in opacity mean determinations; furthermore, radiative accelerations are also a handy byproduct. We therefore designed the OPserver to run on a powerful computer with the whole volume of monochromatic opacities (a few GB) always residing in main memory, whence requests for opacity means could then be resolved relatively fast. The code was developed on an SGI midsize supercomputer at CeCalCULA, Mérida, Venezuela, and finally installed on a dedicated node at the Ohio Supercomputer Center (OSC) coupled with their web server as a front-end (see Figure 4).

Three access modes were considered: (A) the OPserver is downloaded to a powerful local workstation including the database of monochromatic opacities (*mono*) and a Fortran routine library (*OPlibrary*) to link the server with the user modeling code; (B) the *OPlibrary* is downloaded locally and the *mono* database is accessed remotely from the OSC; and (C) the OPserver is accessed through an interactive web page. Mode A has proven to be the most popular as workstation capabilities have grown rapidly and users frequently adapt the *mono* database to fit observed spectra. Mode B was tailored for grid and cloud computing, where the Internet transfer of voluminous datasets is cumbersome while Mode C is for the occasional user who can easily download concise files of mean opacities and radiative accelerations for a handful of chemical mixtures. Modes A and B were designed with heavy calculations of stellar structure or evolution in mind, where mean opacities must be determined at each radial point or time interval, and have not as yet been fully exploited.

---

11 http://cdsweb.u-strasbg.fr/tipbase/home.html.
12 https://www.chiantidatabase.org/.
13 http://opacities.osc.edu/.

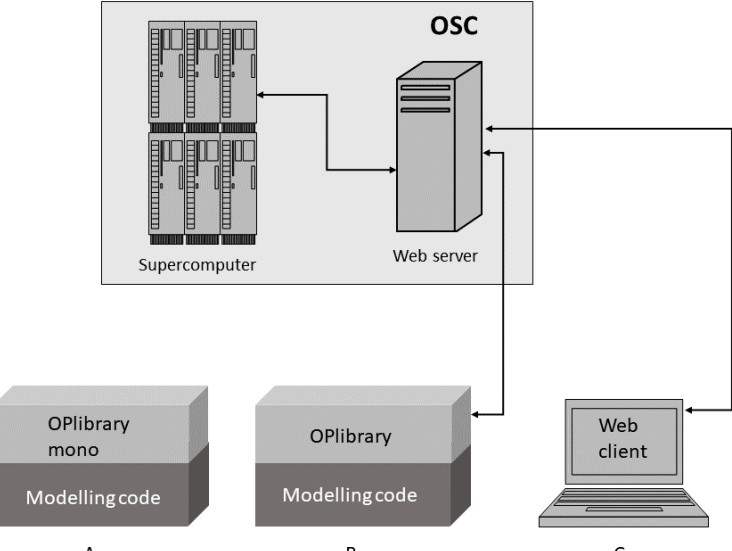

**Figure 4.** OPserver data-service model based at the Ohio Supercomputer Center (OSC) showing its three access modes: (**A**) Local mode where the *OPlibrary* and monochromatic opacities (*mono*) are downloaded and linked to a modeling code; (**B**) Cloud model where the *OPlibrary* is downloaded locally and linked to the modeling code but *mono* is accessed remotely from the OSC; and (**C**): web-page mode. Reproduced from Figure 2 of [33].

The OPserver network programming was carried out originally with a socket interface, which although still operable is now somewhat dated in the realm of web services mostly using HTML or XML application programming interfaces (API) such as the Representational State Transfer (REST) or the Simple Object Access Protocol (SOAP). Since most legacy atomic databases are managed under the relational model, the introduction of XML schemata for data exchange (see, for instance, VAMDC-XSAMS[14]) has revived the hierarchical model. This has given rise to a dichotomy that has not helped database maintenance and upgrading since XML is practically alien to both atomic data producers and users. Moreover, the standardization of the FITS[15] format in astronomical data reservoirs has brought to the table new important considerations.

## 5. AtomPy

AtomPy [34] was a prototype for a cloud-computing environment to promote community-driven curation of atomic data for astrophysical applications, specifically data assessment and preservation. A prospective user is encouraged to not only search for data but also to contribute with datasets for comparison, assessment, and ultimately, preservation. Since the early days (mid 1960s) at the NBS (now NIST), atomic data assessment has been a long-term activity involving dedicated groups of experts elaborating critically evaluated compilations [35–65]. (I quote here a long list of references to exalt the extent of this seminal and long-standing work). In spite of its relevance, sustainable atomic data assessment is nowadays compromised by the contemporary scientific funding time scales that are mostly short-term and project-based. AtomPy hence proposed a self-sustainable model based on an open virtual research community of both atomic data producers and users and on a community-driven data curation model similar to Wikipedia[16]. As discussed in [34], the development of modern data

---

14   https://standards.vamdc.eu/dataModel/vamdcxsams/.
15   https://fits.gsfc.nasa.gov/.
16   https://www.wikipedia.org/.

repositories favors curation procedures that start early in the research cycle and include the data users as well.

The AtomPy[17] atomic data and metadata are stored in spreadsheets in Google Drive where they can be openly accessed, modified, and downloaded. Data downloading for further manipulation is performed through the options offered by Google Sheets[18] or, alternatively, local Pandas DataFrames[19]. Data uploading by prospective contributors to existing or new spreadsheets is at present only carried out through the Google-Drive channels. Data producers, users, and assessors are encouraged to implement data processing workflows in Jupyter Notebooks to be deposited for general use in the AtomPy GitHub repository[20]. Some technical difficulties were encountered with the data volume limit of the Google Sheets and the slow data conversion to Pandas DataFrames, but with time most of these limitations have been surpassed.

As part of the AtomPy 02_02 workbook for He I, Figure 5 shows the 02_02.E0 worksheet containing a 49-level atomic model; it may be seen it lists both the spectroscopic and theoretical level energies (cm$^{-1}$), the reference sources being specified with active ADS links. Following the TOPbase nomenclature, the ionic species are identified with the $(Z, N)$ duplet, where $Z$ and $N$ are respectively the atomic and electron numbers, and in addition to its electron configuration and spectroscopic term, each level is identified with the $(2S + 1, L, Pi, J)$ tuple, wherein $(2S + 1)$ is the spin multiplicity, $L$ the total orbital angular momentum quantum number, $Pi$ the parity, and $J$ the total angular momentum quantum number. Figure 6 depicts the 02_02.A0 worksheet listing $A$-values for the $(Z, N, k, i)$ transitions, wavelengths being obtained from 02_02.E0 worksheet and $A$-values from four different sources. A salient feature in this table is the several empty cells it contains that can indeed be handled by the `pandas.DataFrame` API.

**Fine-Structure Energy Levels for He I**

S1: Spectroscopic energy levels by Kramida, A.E.; Ralchenko, Yu.; Reader, J.; and NIST ASD Team (2013)
http://www.nist.gov/pml/data/asd.cfm
S2: Atomic structure calculations using MCHF and BSR. Zatsarinny, O.; Froese Fischer, C.
http://adsabs.harvard.edu/abs/2009CoPhC.180.2041Z

| Z | N | i | Conf | Term | 2S+1 | L | Pi | J | S1 E(cm-1) | S2 E(cm-1) |
|---|---|---|------|------|------|---|----|---|-----------|-----------|
| 2 | 2 | 1 | 1s2 | 1S | 1 | 0 | 0 | 0.0 | 0.0000000 | 0.0 |
| 2 | 2 | 2 | 1s.2s | 3S | 3 | 0 | 0 | 1.0 | 159855.9743297 | 159831.0 |
| 2 | 2 | 3 | 1s.2s | 1S | 1 | 0 | 0 | 0.0 | 166277.4401410 | 166259.0 |
| 2 | 2 | 4 | 1s.2p | 3P* | 3 | 1 | 1 | 2.0 | 169086.7664725 | 169064.9 |
| 2 | 2 | 5 | 1s.2p | 3P* | 3 | 1 | 1 | 1.0 | 169086.8428979 | 169065.0 |
| 2 | 2 | 6 | 1s.2p | 3P* | 3 | 1 | 1 | 0.0 | 169087.8308131 | 169066.0 |
| 2 | 2 | 7 | 1s.2p | 1P* | 1 | 1 | 1 | 1.0 | 171134.8969460 | 171113.8 |
| 2 | 2 | 8 | 1s.3s | 3S | 3 | 0 | 0 | 1.0 | 183236.7917000 | 183216.9 |
| 2 | 2 | 9 | 1s.3s | 1S | 1 | 0 | 0 | 0.0 | 184864.8293200 | 184848.0 |
| 2 | 2 | 10 | 1s.3p | 3P* | 3 | 1 | 1 | 2.0 | 185564.5619200 | 185546.5 |
| 2 | 2 | 11 | 1s.3p | 3P* | 3 | 1 | 1 | 1.0 | 185564.5838950 | 185546.5 |
| 2 | 2 | 12 | 1s.3p | 3P* | 3 | 1 | 1 | 0.0 | 185564.8545400 | 185546.8 |
| 2 | 2 | 13 | 1s.3d | 3D | 3 | 2 | 0 | 3.0 | 186101.5461767 | 186084.7 |
| 2 | 2 | 14 | 1s.3d | 3D | 3 | 2 | 0 | 2.0 | 186101.5486891 | 186084.7 |
| 2 | 2 | 15 | 1s.3d | 3D | 3 | 2 | 0 | 1.0 | 186101.5928903 | 186084.7 |

**Figure 5.** Workbook 02_02 for He I showing the 02_02.E0 worksheet with a 49-level atomic model (only the first 15 levels are shown).

---

17　http://bit.ly/K5oAfD.
18　https://www.google.com/sheets/about/.
19　https://pandas.pydata.org/pandas-docs/stable/reference/api/pandas.DataFrame.html.
20　https://github.com/AtomPy/AtomPy.

**A-values for fine-structure transitions in He I**

S4: comprehensive tabulation by Wiese, W.L.; Fuhr, J.R.
http://adsabs.harvard.edu/abs/2009JPCRD..38..565W
S2: Atomic structure calculations using MCHF and BSR. Zatsarinny, O.; Froese Fischer, C.
http://adsabs.harvard.edu/abs/2009CoPhC.180.2041Z
S5: Relativistic calculations by Morton, D.C.; Drake, G.W.F.
http://adsabs.harvard.edu/abs/2011PhRvA..83d2503M
and Morton, D.C.; Moffatt, P.; Drake, G.W.F.
http://adsabs.harvard.edu/abs/2011CaJPh..89..129M
S8: Calculation with correlated variational wave functions of the Hylleraas type. Drake, G.W.F.
http://adsabs.harvard.edu/abs/1986PhRvA..34.2871D

Level indices relative to E0

| | | | | | S8 | S5 | | S4 | | | | S2 |
| --- | --- | --- | --- | --- | --- | --- | --- | --- | --- | --- | --- | --- |
| Z | N | k | i | WLVac (A) | A2E1 (s-1) | AE1 (s-1) | AM1 (s-1) | AE1 (s-1) | AE2 (s-1) | AM1 (s-1) | AM2 (s-1) | AE1 (s-1) |
| 2 | 2 | 2 | 1 | 6.25563E+02 | | | | | | 1.2720E-04 | | |
| 2 | 2 | 3 | 1 | | 5.09440E+01 | | | | | | | |
| 2 | 2 | 4 | 1 | 5.91412E+02 | | | | | | | 3.2700E-01 | |
| 2 | 2 | 4 | 2 | 1.08333E+04 | | 1.0213E+07 | | 1.0216E+07 | | | | 1.0222E+07 |
| 2 | 2 | 5 | 1 | 5.91412E+02 | | 1.7758E+02 | | 1.7640E+02 | | | | 1.7306E+02 |
| 2 | 2 | 5 | 2 | 1.08332E+04 | | 1.0213E+07 | | 1.0216E+07 | | | | 1.0222E+07 |
| 2 | 2 | 5 | 3 | 3.55948E+04 | | 2.6897E-02 | | 2.9660E-02 | | | | 2.6910E-02 |
| 2 | 2 | 6 | 2 | 1.08321E+04 | | 1.0213E+07 | | 1.0216E+07 | | | | 1.0225E+07 |
| 2 | 2 | 7 | 1 | 5.84334E+02 | | 1.7983E+09 | | 1.7989E+09 | | | | 1.7967E+09 |
| 2 | 2 | 7 | 2 | 8.86610E+03 | | 1.5489E+00 | | 1.4420E+00 | | | | 1.5467E+00 |
| 2 | 2 | 7 | 3 | 2.05869E+04 | | 1.9749E+06 | | 1.9746E+06 | | | | 1.9734E+06 |

**Figure 6.** Workbook 02_02 for He I showing the 02_02.A0 worksheet with *A*-values for transitions with upper level $k \leq 7$.

## 6. Discussion

Due to the large data volumes involved in the calculation of atomic data for astrophysical applications and to rapidly evolving ICT, data producers have been compelled to develop online databases to facilitate data dissemination. Such ancillary activities involve the mastering of data engineering methods associated with a new way of doing science commonly referred to nowadays as "e-science", which mostly relies on database-centric rather than CPU-centric computing. The four database projects presented here recounts a lengthy learning process along this route.

Apart from the two relevant points in database development of addressing specific data demands in a scientific community and providing online data services of reference, database functionality is key to client acceptance. In other words, a prospective database should be devised more as a data application than a data repository. This aspect favors spectral modeling codes such as CHIANTI [22], CLOUDY [66], XSTAR (this Special Issue), and PyNeb (this Special Issue) that include atomic databases benchmarked with spectral observations. Furthermore, as database volumes are expected to grow dramatically in the current era making data downloading untractable, most modeling activities are being moved to the database end in a cloud environment.

Sustainable atomic data projects are in general compromised by the transient nature of scientific funding and by the rapid evolution of ICT that usually implies regular investment. We discussed the difficulties in database maintenance caused by the changing methods of data exchange, e.g., VAMDC-XSAMS and FITS, and their impact on database structuring. Further points to be considered are metadata management and data provenance and preservation. Attempts to charge a fee for data downloading do not seem to flourish in the present open-data era; thus, in my opinion the funding agencies will have to eventually address this issue in earnest.

We brought to the fore the important issue of atomic data assessment, which until not long ago was mostly carried out by dedicated groups such as that at NIST. Due to recent reorganization in this institution, this important activity may not longer be supported thus opening the field to new practical alternatives. Since data assessment now goes hand in hand with data curation and preservation, we proposed a new scheme (see Section 5) based on an open virtual research community that includes both data producers and users. Initiatives such as this would need further consideration.

It is relevant within the present discussion to say a few words about VAMDC since I was a member of this project and TOPbase, in spite of its advanced age, is included in its current database

registry. VAMDC federates more than 30 diverse atomic and molecular databases and has made serious attempts to make them interoperable, an outstanding shortcoming of the present data infrastructure. VAMDC was a major contributor to the specifications and implementation of the VAMDC-XSAMS XML schema, and adopted it as the standard data exchange protocol. Although apparently correct, this decision opens the door to a rapidly evolving computational maze where no standard seems to prevail. Despite the popularity of HTML, parsing and validating XML schemata can be costly and difficult to manage by both data providers and users apart from the increased data volume due to its trees of data tags. Web-page developers have found Javascript JSON simpler and lighter for data exchange and is becoming a format of choice; however, the fairly old (1980s) FITS format has been formally adopted for image exchange in the upcoming James Webb Space Telescope, and is therefore likely to reinforce its current standardization in astronomy. The comments hereby made regarding database functionality as a key factor for database adoption would also apply to the VAMDC deployment strategy since it concentrates on data fluidity in the distributed application layer rather than developing an application base.

I would conclude by mentioning that the four databases reviewed in this report are still accessible although they are not dynamically updated as they contain data associated with specific projects. They have nevertheless undergone technical upgrades; for instance, web user interfaces and, as previously mentioned, TOPbase was migrated to a MySQL DBMS. The OPserver has been recently adopted as a study case at the OSC for container deployment.

**Funding:** The work for this review was carried out while I was a recipient of a grant from the NASA Astrophysics Research and Analysis Program (grant 80NSSC17K0345).

**Acknowledgments:** I am indebted to Mike Seaton (UCL), my PhD thesis supervisor and mentor, for guidance and encouragement in my first steps in data science. I am also grateful to Claude J. Zeippen (Observatoire de Paris, France), Anil K. Pradhan (Ohio State University, USA), Walter Cunto (then at IBM Venezuela Scientific Center, Caracas, Venezuela), François Ochsenbein (CDS, Strasbourg, France), Luis A. Núñez (then at CeCalCULA, Mérida, Venezuela, now at the Universidad Industrial de Santander, Bucaramanga, Colombia), Marcio Meléndez (then at Universidad Simón Bolívar, Caracas, now at the Space Telescope Science Institute, Baltimore, USA), Manuel A. Bautista (then at IVIC now at Western Michigan University, USA), Timothy R. Kallman (NASA Goddard Space Flight Center, USA), Javier A. García (then at IVIC now at Caltech, USA), and Juan González (then at Universidad de Carabobo, Valencia, Venezuela, now at Amazon, Madrid, Spain) for productive collaborations and support in the project management and implementation of the atomic databases reviewed in this report. I deeply thank the CDS and OSC for reliable and long-term (several decades) database hosting and technical support.

**Conflicts of Interest:** The author declares no conflict of interest.

## Abbreviations

The following abbreviations are used in this manuscript:

| | |
|---|---|
| ADS | Astrophysics Data System |
| API | Application Programming Interface |
| DBMS | Database Management System |
| CeCalCULA | Centro de Cálculo Científico Universidad de Los Andes |
| CDS | Centre de Données astronomiques de Strasbourg |
| CPU | Central Processing Unit |
| FITS | Flexible Image Transport System |
| IAU | International Astronomical Union |
| ICT | Information and Communications Technologies |
| IDL | Interactive Data Language, a product of Harris Geospatial Solutions |
| IP | Iron Project |
| NBS | National Bureau of Standards |
| NIST | National Institute of Standards and Technology |
| OP | Opacity Project |
| OPAL | Opacities at Livermore |
| OSC | Ohio Supercomputer Center |
| OSU | Ohio State University |

REST          Representational State Transfer
SDSS          Sloan Digital Sky Survey
SGI           Silicon Graphics Inc.
SOAP          Simple Object Access Protocol
UCL           University College London
VAMDC         Virtual Atomic and Molecular Data Center
XSAMS         XML Schema for Atomic, Molecular and Solid Data

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
