# Peer review of "Atomic Databases: Four of a Kind"

_atoms, doi:10.3390/atoms8020030_

Round 1

Reviewer 1 Report

Review of”Atomic Databases: Four of a Kind“ by Claudio Mendoza.
This paper describes four atomic databases, starting with one that existed initially as tables in a paper by the author, then progressing to TIPTOPbase, Observer, and AtomPy. The paper describes the motivation for each database, a brief review of the atomic data and storage/retrieval methods for each, and their current status. It concludes with a discussion of issues faced by all such databases and some recommendations for the future.

This paper is well-written, concise but complete, and makes good use of figures. I include below a number of suggestions to the author, but consider none of them essential; I would judge the paper ready for acceptance.

My suggestions include:

1) The paper notes at the end of Section 2) that a number of attempts to create more complete atomic databases for nebular emission did not reach popularity until the creation of CHIANTI in the late 1990s. Since any attempt to create an atomic database requires substantial work, and the author has experience in this area, it would be a useful addition to speculate why this was the case. Were the intervening attempts too small an increase to justify switching? Were they too hard to use, or were they not well advertised? CHIANTI is a major endeavor whose use goes well beyond modeling nebular emission, so its success is perhaps not unexpected, but reasons for the non-use of other efforts would be of interest.

2) In Section 3), line #93, the paper noted that the OP was a pioneer of collaborative big-data science, an undoubtedly true statement. It would be useful to comment briefly on some of the details of this arrangement. Was it the vision of one person, was it organized via a joint conference, how was it managed, etc? Given the size and complexity of modern atomic databases, this kind of collaboration will be essential and some insight into what made the OP work would be helpful.

3) In Section 3), line #138, the paper notes without comment, "As the OP came to an end." Given that the section opened noting that the OP was in response to a `plea', it would be of interest to know if the plea was answered (and thus the OP end), or if the OP ended for some other reason.

4) In Section 4), the paper notes that users of the OPserver primarily use Mode A of access as "workstation capabilities have grown rapidly." Unsurprisingly, this is a common theme for atomic databases from the OP on - the initial size exceeds current computing facilities and substantial work is put into addressing these limitations, but then computers leap ahead. It seems that AtomPy (Section 5) might be the only system not designed to work around computing limitations; is this true? The paper does note in Section 6) that modeling will have to be done in a cloud environment due to database size, suggesting that yet again developers will need to use new techniques.

5) At the end of Section 4), the paper notes that the OPserver's network access model is now dated as new approaches have become popular. Given the transient nature of funding for such projects, and the rapid pace of change in the computing field, this will continually occur. It would be a useful addition if the author could make any recommendations to limit or avoid these problems.

6) The paper mentions the VAMDC (line 110) briefly, but does not mention it in the Discussion (Section 6), focusing instead on tools like CHIANTI, CLOUDY, XSTAR, and PyNeb as "data applications" rather than a "data repository." The VAMDC is one model of a modern cloud-accessed suite of databases (see point 4 above), so some comment would be appropriate here. Is the VAMDC simply ahead of its time, or, in the author's opinion, is there some flaw in the design or implementation that has led to its slow acceptance by the community. Without a doubt the four tools mentioned get far more use and reference than VAMDC, so this is not an unfair assessment.

Author Response

I would like to thank the referee for raising important points that have led to an improved manuscript. The addenda are included in the text with bold face, and here I explain the actions resulting from the referee's comments.

1. The paper notes at the end of Section 2) that a number of attempts to create more complete atomic databases for nebular emission did not reach popularity until the creation of CHIANTI in the late 1990s. Since any attempt to create an atomic database requires substantial work, and the author has experience in this area, it would be a useful addition to speculate why this was the case. Were the intervening attempts too small an increase to justify switching? Were they too hard to use, or were they not well advertised? CHIANTI is a major endeavor whose use goes well beyond modeling nebular emission, so its success is perhaps not unexpected, but reasons for the non-use of other efforts would be of interest.

REPLY: In line 73 I included the following sentence: “In opinion this was due to slow piecemeal improvements and the reluctance of users to replace standard reference data. The widely shared atomic database allowed the nebular modeling community to concentrate on the astrophysics rather than on the uncertainties of the underlying physical data.”

2.  In Section 3), line #93, the paper noted that the OP was a pioneer of collaborative big-data science, an undoubtedly true statement. It would be useful to comment briefly on some of the details of this arrangement. Was it the vision of one person, was it organized via a joint conference, how was it managed, etc? Given the size and complexity of modern atomic databases, this kind of collaboration will be essential and some insight into what made the OP work would be helpful.

REPLY: In line 99 I included a new paragraph: "The workload was divided into isoelectronic sequences that were assigned to the respective research groups. Progress was monitored on a six-monthly basis in OP meetings held in the different participating countries and, in the latter stages of the project, through an email list. The atomic data compilation by means of half-inch magnetic tapes and exabyte cartridges was coordinated by Mike Seaton himself, who devised a series of utilities to test data integrity leading in several cases to recalculations or new calculations (e.g. the PLUS-data [26]). I implemented a second set of tests for TOPbase mainly concerned with term assignments and spectroscopic series accuracy and completeness. Publication was carried out in series of papers (``The equation of state for stellar envelopes'', ``Atomic data for opacity calculations'') and in two books [27]."

3. In Section 3), line #138, the paper notes without comment, "As the OP came to an end." Given that the section opened noting that the OP was in response to a `plea', it would be of interest to know if the plea was answered (and thus the OP end), or if the OP ended for some other reason.

REPLY: The phrase "As the OP came to an end," has been made more precise being replaced in line 153 by “As the OP computations of the atomic radiative data came to an end in the mid-1990s.” The OP in fact carried on until the mid-2000s but mainly concerned with opacity tables.

4. In Section 4), the paper notes that users of the OPserver primarily use Mode A of access as "workstation capabilities have grown rapidly." Unsurprisingly, this is a common theme for atomic databases from the OP on - the initial size exceeds current computing facilities and substantial work is put into addressing these limitations, but then computers leap ahead. It seems that AtomPy (Section 5) might be the only system not designed to work around computing limitations; is this true? The paper does note in Section 6) that modeling will have to be done in a cloud environment due to database size, suggesting that yet again developers will need to use new techniques.

REPLY: Not quite, computer limitations are always there. In line 229 I have included the phrase: “Some technical difficulties were encountered with the data volume limit of the Google Sheets and the slow data conversion to Pandas DataFrames, but with time most of these limitations have been surpassed.”

5. At the end of Section 4), the paper notes that the OPserver's network access model is now dated as new approaches have become popular. Given the transient nature of funding for such projects, and the rapid pace of change in the computing field, this will continually occur. It would be a useful addition if the author could make any recommendations to limit or avoid these problems.

REPLY: this is a very important point for which I am grateful. I have thus introduced in line 203 the text: “Since most legacy atomic databases are managed under the relational model, the introduction of XML schemata for data interchange (see, for instance, VAMDC-XSAMS13) has revived the hierarchical model. This gives rise to a dichotomy that has not helped database maintenance and upgrading since XML is practically alien to both atomic data producers and users. Moreover, the standardization of the FITS format in astronomical data reservoirs has brought to the table new important considerations.” I also included in Section 6 (line 258) the following paragraph: “Sustainable atomic data projects are in general compromised by the transient nature of scientific funding and by the rapid evolution of ICT that usually implies regular investment. We discussed the difficulties in database maintenance caused by the changing methods of data interchange, e.g. XSAMS and FITS, and their impact on database structuring. Further points to be considered are metadata management and data provenance and preservation. Attempts to charge a fee for data downloading do not seem to flourish in the present open-data era; thus, in my opinion the funding agencies will have to eventually address this issue in earnest.”

6. The paper mentions the VAMDC (line 110) briefly, but does not mention it in the Discussion (Section 6), focusing instead on tools like CHIANTI, CLOUDY, XSTAR, and PyNeb as "data applications" rather than a "data repository." The VAMDC is one model of a modern cloud-accessed suite of databases (see point 4 above), so some comment would be appropriate here. Is the VAMDC simply ahead of its time, or, in the author's opinion, is there some flaw in the design or implementation that has led to its slow acceptance by the community. Without a doubt the four tools mentioned get far more use and reference than VAMDC, so this is not an unfair assessment.

REPLY: Yes, the article was really missing some more comments on VAMDC since TOPbase is part of its registry. At the end of the Section 6 (line 271) I have included a paragraph on VAMDC: "It is relevant within the present discussion to say a few words about VAMDC since I was a member of this project and TOPbase, in spite of its advanced age, is included in its current database registry. VAMDC federates more than 30 diverse atomic and molecular databases and has made serious attempts to make them interoperable, an outstanding shortcoming of the present data infrastructure. VAMDC was a major contributor to the specifications and implementation of the VAMDC-XSAMS XML schema and adopted it as the standard data exchange protocol. Although apparently correct, this decision opens the door to a rapidly evolving computational maze where no standard seems to prevail. Despite the popularity of HTML, parsing and validating XML schemata can be costly and difficult to manage by both data providers and users apart from the increased data volume due to its trees of data tags. Web-page developers have found Javascript JSON simpler and lighter for data exchange and is becoming a format of choice; however, the fairly old (1980s) FITS format has been formally adopted for image exchange in the upcoming James Webb Space Telescope, and is therefore likely to reinforce its current standardization in astronomy. The comments hereby made regarding database functionality as a key factor for database adoption would also apply to the VAMDC deployment strategy as it mainly focus data fluidity in the distributed application layer rather than developing an application base.”

Reviewer 2 Report

I am assuming that this paper is intended as a contribution to the upcoming Special Issue Development and Perspectives of Atomic and
Molecular Databases, possibly the opening paper in the issue, given that
the author is also the guest editor of the Special Issue.  That would
explain and perhaps justify the somewhat autobiographical sentences at
the beginning of Sections 1 and 2 and also the Acknowledgements which
tend towards the self-congratulatory.  Normally these would be rephrased for a more usual research paper, but I do not suggest changing them in the present context.

The paper includes a description of four databases, and to some extent
the differences between them.  The Discussion section is rather minimal.
What I think would be useful is an additional, short
Conclusions/Summary section in which the pointers forward into the
future are summarised.  It would take the reader quite a bit of digging
into the text of the paper to draw out this important guidance.  It
would also provide a better focus of the purpose of the paper which is
somewhat lacking without such digging.

On a few small matters,

(a) I suggest changing the word 'ancient' (line 18 of section 2) to
'early'.  'ancient' suggests more digging - archaeological!

(b) The use of the term 'best-seller' (line 21 of section 2) is a rather
extravagant claim.  I would suggest replacing it by 'highly-cited'.

(c) Section 3 is headed 'TIPTOPbase.  The section talks quite a bit
about TOPbase, which contains data from the Opacity Project, and about
TIPbase, containing data from the Iron Project.  But TIPTOPbase isn't
actually defined.  Is it the merging of TOPbase and TIPbase?  Please
provide a brief explanation.

Author Response

I thank the referee for the comments, which have been taken into consideration. Addenda in the text have been made in bold face, and here I give account of the actions made.

The paper includes a description of four databases, and to some extent
the differences between them.  The Discussion section is rather minimal.
What I think would be useful is an additional, short
Conclusions/Summary section in which the pointers forward into the
future are summarised.  It would take the reader quite a bit of digging
into the text of the paper to draw out this important guidance.  It
would also provide a better focus of the purpose of the paper which is
somewhat lacking without such digging.

REPLY: as requested by the other two referees, Section 6 (Discussion) has been substantially extended, and in my opinion, covers the points suggested by the referee.

On a few small matters,

(a) I suggest changing the word 'ancient' (line 18 of section 2) to 'early'.  'ancient' suggests more digging - archaeological!

REPLY: the suggested change has been made.

(b) The use of the term 'best-seller' (line 21 of section 2) is a rather
extravagant claim.  I would suggest replacing it by 'highly-cited'.

REPLY: the phrase “to become a best seller and my most cited paper ever” has been replaced by “to become a highly cited paper”.

(c) Section 3 is headed 'TIPTOPbase.  The section talks quite a bit
about TOPbase, which contains data from the Opacity Project, and about
TIPbase, containing data from the Iron Project.  But TIPTOPbase isn't
actually defined.  Is it the merging of TOPbase and TIPbase?  Please
provide a brief explanation.

REPLY: Section 3 now begins with the paragraph: “The term TIPTOPbase -- alluding to the adjective “tip-top” for the very best class and quality -- refers to the TOPbase and TIPbase atomic databases of the Opacity Project (OP) and Iron Project (IP).”

Reviewer 3 Report

Dear Claudio,

I find the paper interesting as a lesson about the learning steps.

About minor changes :

 - Have a look at sentence "The access to powerful .. " from line 48 to 51. I think that it would help to improve the structure of that sentence

Then more scientifically, I would suggest 3 specific points to address :

  - About VAMDC implementation of TOPbase : i understood from Nicolas Moreau that they transfered the database to a mysql database - do you feel like contacting him to include more infos in the paper ? if not, this is not a pb

  - About TIPbase : I kind of understand that the 2 comments from line 146 to line 152 are cristisms of TIPbase. It would help to understand why CHIANTI is better with respect to the 2nd comment (provenance of data). In any case the fate of TIPbase and the comments could be more related

In general for all databases, applications that you mention : I believe that it would help the readers to know their current status with respect to the following items :

  • are they still accessible
  • are they archives or dynamically scientifically updated
  • are there any technical maintenance or upgrades

All the best

 -

Author Response

I am very grateful for the comments and have made changes to the manuscript as suggested. Addenda in the text are made in bold face, and here I give an account of the actions taken.

About minor changes :

 - Have a look at sentence "The access to powerful .. " from line 48 to 51. I think that it would help to improve the structure of that sentence

REPLY: this sentence has been rewritten in line 48: “The access to powerful computers in the 1970s led to a new generation of structure and electron–ion scattering calculations, which took into account electron correlation effects formally by the configuration interaction [8–10] and close-coupling [11–14] methods. Relativistic effects were introduced algebraically, with the Breit–Pauli Hamiltonian, or with the fully relativistic Dirac Hamiltonian.”

Then more scientifically, I would suggest 3 specific points to address :

  - About VAMDC implementation of TOPbase : i understood from Nicolas Moreau that they transfered the database to a mysql database - do you feel like contacting him to include more infos in the paper ? if not, this is not a pb

 REPLY: You are right, a transfer rather than a transcription. The sentence has been replaced in line 122 by the more precise, “It is worth mentioning that the original TOPbase at the CDS is still operable, but was recently transferred to a MySQL DBMS by Franck Delahaye and Nicolas Moreau (Observatoire de Paris, France) to integrate it to the portal of the Virtual Atomic and Molecular Data Center (VAMDC), an ambitious European project to integrate several (more than 30) atomic and molecular databases [30,31].”

  - About TIPbase : I kind of understand that the 2 comments from line 146 to line 152 are cristisms of TIPbase. It would help to understand why CHIANTI is better with respect to the 2nd comment (provenance of data). In any case the fate of TIPbase and the comments could be more related

REPLY: The following sentence was added in line 167: “TIPbase was conceived only to display data computed in the IP and could not then compete in completeness with CHIANTI that compiled data from different sources.”

In general for all databases, applications that you mention : I believe that it would help the readers to know their current status with respect to the following items :

  • are they still accessible
  • are they archives or dynamically scientifically updated
  • are there any technical maintenance or upgrades

REPLY: An additional paragraph was inserted at the end of Section 6 (line 287): "I would conclude by mentioning that the four databases reviewed in this report are still accessible although they are not dynamically updated as they contain data associated with specific projects. They have nevertheless undergone technical upgrades; for instance, web user interfaces and, as previously mentioned, TOPbase was migrated to a MySQL DBMS. The OPserver has been
recently adopted as a study case at the OSC for container deployment."